

# CompLaB v1.0: a scalable pore-scale model for flow, biogeochemistry, microbial metabolism, and biofilm dynamics

Heewon Jung[1,2], Hyun-Seob Song[3], Christof Meile[1]

[1]Department of Marine Sciences, University of Georgia, Athens, GA 30602, USA

[2]Department of Geological Sciences, Chungnam National University, Daejeon 34134, South Korea

[3]Department of Biological Systems Engineering, Department of Food Science and Technology, Nebraska Food for Health Center, University of Nebraska-Lincoln, Lincoln, NE 68583, USA

*Correspondence to*: Christof Meile (cmeile@uga.edu)

**Abstract.** Microbial activity and chemical reactions in porous media depend on the local conditions at the pore scale and can

involve complex feedback with fluid flow and mass transport. We present a modeling framework that quantitatively accounts for the interactions between the bio(geo)chemical and physical processes, and that can integrate genome-scale microbial metabolic information into a dynamically changing, spatially explicit representation of environmental conditions. The model couples a Lattice-Boltzmann implementation of Navier-Stokes (flow) and advection-diffusion-reaction (mass conservation) equations. Reaction formulations can include both kinetic rate expressions and flux balance analyses, thereby integrating

reactive transport modeling and systems biology. We also show that the use of surrogate models such as neural network representations of in silico cell models can speed up computations significantly, facilitating applications to complex environmental systems. Parallelization enables simulations that resolve heterogeneity at multiple scales, and a cellular automata module provides additional capabilities to simulate biofilm dynamics. The code thus constitutes a platform suitable for a range of environmental, engineering and – potentially - medical applications, in particular ones that involve the simulation

of microbial dynamics.

## 1 Introduction

Biogeochemical turnover in Earth's near-surface environments is governed by the activity of microbes adapted to their surroundings to catalyze reactions and gain energy. In turn, these activities shape the environmental composition, which feeds back on metabolic activities and creates ecological niches. Such feedbacks can be captured by reactive transport models that

compute the evolution of geochemical conditions as a function of time and space and simulate microbial activities in porous media (Meile and Scheibe, 2019). Commonly used macroscopic reactive transport models simplify small scale features of natural porous media. For example, heterogeneous pore geometry and transport phenomena are represented by only a few macroscopic parameters such as porosity, permeability, and dispersivity (Steefel et al., 2015). However, such simplifications can lead to a disparity between model estimations and actual observations because these models do not resolve the physical

and geochemical conditions at the scale that is relevant for microbial activity (e.g., Molins 2015, Oostrom et al. 2016).



Furthermore, microbial reaction rates are often formulated using Monod expressions, which describe a dependency of metabolic rates on nutrient availability, but substantially simplify the complex metabolic adaptation of microbes in changing environments. This recognition has prompted the development of constraint-based models including, for example, COMETS (Harcombe et al., 2014), BacArena (Bauer et al., 2017), and IndiMeSH (Borer et al., 2019), which have enabled detailed

descriptions of complex microbial metabolisms and metabolic interactions (Dukovski et al. 2021). However, most constraint-based models are not designed to capture combined diffusive and advective transport of metabolites in heterogeneous subsurface environments and are not optimized to handle such settings in a computationally efficient way.

To account for the feedback between environmental conditions, chemical processes, microbial metabolism, and structural changes in the porous medium caused by these activities, we introduce a novel pore scale reactive transport modeling

framework with spatially explicit descriptions of hydrological and biogeochemical processes. The framework is developed to account for various chemical reactions and/or genome-scale metabolic models with advective and diffusive transports in porous media at the pore scale. The Lattice Boltzmann (LB) method is used to compute fluid flow and solute transport in complex porous media, capable of simulating both advection and diffusion dominated settings. Microbial metabolism and chemical reactions are incorporated as source or sink terms in the LB method solving mass conservation equations. These

sources or sinks can be described classically using approximations such as Monod kinetics (Tang et al., 2013), or can be derived from cell scale growth and metabolic fluxes simulated with flux balance analysis (Orth et al. 2010). Biomass dynamics can be described by keeping track of cell densities (similar to chemical concentration fields) of different organisms or populations, with cell movement either based on an advection-diffusion formulation or using a cellular automata approach. In addition, we incorporate a surrogate modeling approach to make larger scale simulations possible. Thus, the framework

provides options either to maximize computational efficiency via the use of surrogate models, or to directly utilize well-established metabolic modeling environments without losing the inherent parallel scalability of the LB method. The model is validated by comparing model simulations to published simulation results. We demonstrate the flexibility of the new microbial reactive transport framework, its scalability, and the benefits of using surrogate models to circumvent computational bottlenecks posed by flux balance analysis. Our work therefore facilitates cross-disciplinary efforts that integrate bioinformatic

approaches underlying cell models with descriptions suitable to resolve the dynamic nature of natural environments. This allows for the representation of microbial interactions, which is a major challenge to our current quantitative understanding of microbially mediated elemental cycling (Sudhakar et al. 2021).

## 2 Use of open-source codes

To establish a modeling framework that builds on the existing and future knowledge and know-how from multiple disciplines,

our approach uses the open-source software Palabos and integrates it with an open-source linear programming solver GLPK, and the COBRApy (CPY) python package for genome-scale metabolic modeling. Palabos (**Pa**rallel **La**ttice **Bo**ltzmann **S**olver) is a modeling platform that has established itself as a powerful approach in the field of computational fluid dynamics based on



the Lattice Boltzmann (LB) method. The Palabos software is designed to be highly extensible to couple complex physics and other advanced algorithms without losing its inherent capability of massive parallelization (Latt et al., 2021). Palabos has been

parallelized using the Message Passing Interface where computational domains are subdivided while minimizing the inter-process communication. It has been used for building modeling platforms to simulate deformable cell suspensions in relation to blood flows (Kotsalos et al., 2019) and complex subsurface biogeochemical processes at the pore scale (Jung and Meile, 2019, 2021). It is highly scalable and hence was chosen as high-performance modeling framework to be integrated with our representations of chemical and microbial dynamics. GLPK (**G**NU **L**inear **P**rograming **K**it) is an open-source library designed

for solving linear programming (LP), mixed integer programming and other related problems (Makhorin, 2009). It contains the simplex method, a well-known efficient numerical approach to solve LP problems, and the interior-point method, which solves large-scale LP problems faster than the simplex method. GLPK provides application programming interface (API) written in C language to interact with a client program. COBRApy is an object-oriented python implementation of **CO**nstraint-**B**ased **R**econstruction and **A**nalysis (COBRA) methods (Ebrahim et al. 2013) which is suitable to be integrated with other

libraries without requiring commercial software. Through a simple python API, the fast evolving and expanding biological modeling capacity of COBRApy, which includes features such as flux balance analysis (FBA), flux variability analysis (FVA), metabolic models (M-models), and metabolism and expression models (ME-models), can be employed.

## 3 Model description

*CompLaB* simulates 2D fluid flow and solute transport based on the LB method implemented in Jung and Meile (2019, 2021).

The LB method is particularly useful for simulating subsurface processes because boundaries between solid and fluid can be handled by a simple bounce-back algorithm (Ziegler, 1993) in addition to its massive parallelization efficiency (Latt et al., 2021). For these reasons, the LB method has been applied to simulate a broad range of pore scale reactive transport processes (e.g., Huber et al., 2014; Kang et al., 2007; Tang et al., 2013). The simulations are guided by an input file, CompLaB.xml, that sets the scope of the simulation and capabilities utilized through command blocks that define the model domain, chemical

state variables, microorganisms, and model input/output (Figure 1). Below, the features of the model are presented, and we refer to the manual in the code repository for examples of the implementation.



**Figure 1.** Flowchart of *CompLaB* for reactive transport simulations.






### 3.1 The Lattice-Boltzmann flow and mass transport solvers

The LB method retrieves the numerical solutions of the Navier-Stokes (NS) and advection-diffusion-reaction equations (ADRE) by solving the mesoscopic Boltzmann equation across a defined set of particles (Krüger et al., 2017). *CompLaB* obtains a steady state flow field by running a flow solver with a D2Q9 lattice


$$f_i(\mathbf{r} + \boldsymbol{c}_i \Delta t, t + \Delta t) = f_i(\mathbf{r}, t) + \Omega_i^{BGK}(\mathbf{r}, t) \tag{1}$$

where $f_i$(r,t) is the $i^{th}$ discrete set of particles streamed from a position r to a new position r+$c_i$ after a time step with lattice velocities $c_i$ ($c_0 = [0,0]$, $c_1 = [1,0]$, $c_2 = [0,1]$, $c_3 = [-1,0]$ , $c_4 = [0,-1]$ , $c_5 = [1,1]$ , $c_6 = [-1,1]$ , $c_7 = [-1,-1]$ , $c_8 = [1,-1]$). $\Omega_i^{BGK}$ is the Bhatnagar-Gross-Krook collision operator (Bhatnagar et al. 1954) which is given by


$$\Omega_i^{BGK}(\mathbf{r}, t) = \frac{\Delta t}{\tau} \left[ \omega_i \rho + \omega_i \rho_0 \left( \frac{\mathbf{u} \cdot \boldsymbol{c}_i}{c_s^2} + \frac{(\mathbf{u} \cdot \boldsymbol{c}_i)^2}{2c_s^4} - \frac{\mathbf{u} \cdot \mathbf{u}}{2c_s^2} \right) - f_i(\mathbf{r}, t) \right] \tag{2}$$

where $\tau$ is the relaxation time, $\omega_i$ are the lattice weights ($\omega_0 = 4/9$, $\omega_{1-4} = 1/9$, $\omega_{5-8} = 1/36$), $\rho$ is the macroscopic density ($\rho = \Sigma$ $f_i$), $\rho_0$ is the rest state constant, u is the macroscopic velocity calculated from the momentum ($\rho$u = $\Sigma$ $c_i f_i$), and $c_s$ is a lattice dependent constant (here, $c_s^2 = 1/3$).

The steady state flow field is then imposed on a transport solver defined as

$$g_i(\mathbf{r} + \boldsymbol{c}_i \Delta t, t + \Delta t) = g_i(\mathbf{r}, t) + \Omega_i^{BGK}(\mathbf{r}, t) + \Omega_i^{RXN}(\mathbf{r}, t) \tag{3}$$

where $g_i(\mathbf{r}, t)$ represents the discrete particle set $i$ at position $r$ and time $t$. A D2Q5 lattice, which satisfying the isotropy requirement for a LB transport solver, is used for numerical efficiency with the lattice weights $\omega_0 = 1/3$ and $\omega_{1-4} = 1/6$, and

lattice velocities $c_{0-4}$. With a steady-state flow field obtained from the solution of Equation 1 and a reaction step $\Omega_i^{RXN} = \Delta t \omega_i R$, the LB transport solver recovers an ADRE with the following form

$$\frac{\partial J}{\partial t} = \nabla \cdot (D_J \nabla J) - \mathbf{u} \cdot \nabla J + R \tag{4}$$

where $J$ is a transported entity, including a solute concentration ($C$) and a planktonic biomass density, with a diffusivity $D_J =$

$c_s^2 \left( \tau_J - \frac{\Delta t}{2} \right)$, and $R$ is a reaction term computed by the reaction solver of *CompLaB* as described below.





### 3.2 Reactions

The reaction step ($\Omega_i^{RXN}$) computation is separated from a transport computation via the sequential non-iterative approach (Alemani et al., 2005). A unique feature of *CompLaB* is that its reaction solver can compute biochemical reaction rates $R$ through (1) kinetic rate expressions, (2) flux balance analysis, (3) a surrogate model such as a pre-trained artificial neural

network, or combinations thereof.

### 3.2.1 Kinetic rate expressions

*CompLaB* provides a C++ template that users can adapt to formulate kinetic rate expressions using metabolite concentrations and biomass densities (`defineReactions.hh`). This is designed to accommodate user-specific needs and to enable simulating various microbial dynamics including Monod kinetics, microbial attachment, and detachment. Reactions can be

restricted to particular locations using material numbers (`mask`) differentiating fluid, biomass and grain surfaces.

Local biomass densities and concentrations calculated after the collision step of transport solvers are transferred to the function as vectors $B$ and $C$ where the vector elements follow the order defined in the user interface (`CompLaB.xml`). The biomass density and metabolite concentrations are updated according to:

$$B_{t+\Delta t} = B_t + \gamma_t B_t \Delta t \tag{5}$$

$$C_{t+\Delta t} = C_t + R_t \Delta t \tag{6}$$

where the cell specific biomass growth rate ($\gamma_t$), and for microbially mediated reactions, the rate $R_t$, is expressed as the product of (user-defined, typically substrate concentration-dependent) metabolite uptake/release rates ($F_t$) per cell multiplied by the cell density $B_t$ ($R_t = F_t B_t$), calculated every timestep for every pore and biomass grid cell.

### 3.2.2 Flux balance analysis

For genome-enabled metabolic modeling, *CompLaB* loads metabolic networks and calculates microbial growth rates as well

as metabolite uptake/release rates through an FBA method (Orth et al., 2010). FBA investigates the metabolic capabilities by imposing several constraints on the metabolic flux distributions. Assuming that metabolic systems are at steady state, the system dynamics for a metabolic network is described by the mass balance equation, $\boldsymbol{Sv} = \boldsymbol{0}$. Here, $\boldsymbol{S}$ is a $m \times n$ matrix with $m$ compounds and $n$ reactions where the entries in each column are the stoichiometric coefficients of the metabolites composing a reaction, and $\boldsymbol{v}$ is a $n \times 1$ flux vector representing metabolic reactions and uptake/release of chemicals by the cell. Most

metabolic models have more reactions than compounds ($n > m$) meaning that there are more unknowns than equations. To solve such underdetermined systems, FBA confines the solutions to a feasible set by imposing constraints on metabolic fluxes $\boldsymbol{lb}$ *(lower bounds)* $\leq \boldsymbol{v} \leq \boldsymbol{ub}$ *(upper bounds)* and applies an objective function $f(\boldsymbol{v}) = \boldsymbol{c}^\mathrm{T}\boldsymbol{v}$, where $\boldsymbol{c}$ is the vector of weights for the objective function to identify an optimal solution. Commonly used objective functions include maximization of biomass yield,





maximization of ATP production, and minimization of nutrient uptake (Nikdel et al. 2018). *CompLaB* utilizes the
stoichiometric matrix **S** from standard metabolic databases such as BiGG and KEGG which are widely used in FBA simulation
environments (e.g., COBRA toolbox (Heirendt et al., 2019), COBRApy (Ebrahim et al., 2013), KBase (Arkin et al., 2018)).
Therefore, *CompLaB* can integrate many existing *in silico* cell models.

   *CompLaB* computes the solution of the metabolic models at each point in space and time for each organism or microbial
community (if the model represents multiple microorganisms) and updates biomass density and metabolite concentrations
according to Equations 5 and 6. The metabolic uptake fluxes are set through imposing constraints by defining the lower bound
(**lb**) of a chemical (uptake fluxes are negative) through one of the following approaches. The first is the parameter-based
method employed by Harcombe et al., (2014), setting the metabolic fluxes in analogy with Michaelis-Menten kinetics using a
maximum uptake rate ($V_{max}$; mmol/g$_{DW}$/h)

$$lb = -V_{max}\left(\frac{C}{C + K_s}\right) \tag{7}$$


where $C$ is a local metabolite concentration (mM), and $K_s$ is a half-saturation constant (mM). The second is the semi-linear
approach employed by Borer et al. (2019). This method replaces $V_{max}$ with $C/B\Delta t$ where $B$ is a local biomass density (g$_{DW}$/L)
and $\Delta t$ is the length of a time step measured in hours (h)

$$lb = -\frac{C}{B\Delta t}\left(\frac{C}{C + K_s}\right) \tag{8}$$


If $K_s$ is set to 0, then the uptake flux estimation becomes a linear function to local concentrations. With lower bounds defined,
the solution of an FBA problem outputs biomass growth rate ($\gamma$, 1/h) and uptake/release rates of metabolites ($F$; mmol/g$_{DW}$/h).

### 3.2.3 Surrogate model

*CompLaB* also provides a C++ template (`surrogateModel.hh`) where users can incorporate a pre-trained surrogate model
for calculating biogeochemical reactions, including artificial neural networks (ANN). This functionality can be used to replace
FBA which requires solving many computationally expensive linear optimization problems (section 3.2.2). In the example
shown in section 5, *CompLaB* provides local metabolite concentrations and biomass densities as inputs and the surrogate model
outputs microbial growth rate ($\gamma$) and uptake/production rates of metabolites ($F$). While our demonstration is based on ANN
models, any pre-trained statistical surrogate model (e.g., De Lucia and Kühn, 2021) that describes the sources and sinks – or
their parameterization – can be used to enhance computational efficiency and accommodate various user-specific needs.





### 3.3 Biomass redistribution

To explicitly model the spatial biomass expansion, *CompLaB* utilizes a cellular automaton (CA) with a predefined maximum biomass density ($B_{max}$) based on the CA algorithm developed by Jung and Meile (2021). After updating local biomass densities, the CA algorithm checks at every time steps if there is any grid cell exceeding $B_{max}$ and redistributes the excess biomass ($B - B_{max}$) to a randomly selected neighboring grid cell. If the selected grid cell cannot hold the excess biomass, the first chosen grid cell is filled up to the maximum holding capacity ($B_{max}$) and then the remaining excess biomass is allocated to a randomly chosen second neighbor cell. If all the neighboring grid cells are saturated with biomass ($B \geq B_{max}$) and hence the excess biomass cannot be placed, the Manhattan distances of biomass grid cells to the closest pore cells are evaluated. The remaining excess biomass is then placed in a neighboring grid cell that is closer to pores and this biomass allocation process is repeated until all the excess biomass is redistributed. Figure 2 shows an example of the cellular automaton process for biomass redistribution.

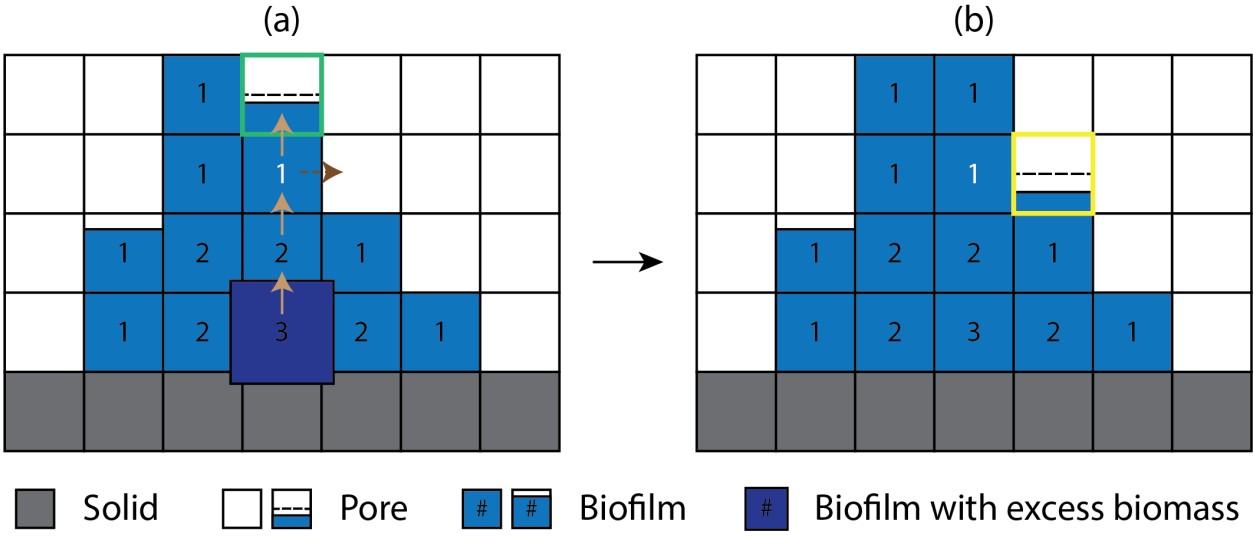

**Figure 2.** Schematic representation of the cellular automaton method for biomass spreading (a) before and (b) after biomass redistribution. The solid, pore, and biomass are color-coded gray, white and blue, respectively. Brown solid arrows indicate the randomly chosen path excess biomass travels. The numbers in the biofilm grid cells are the Manhattan distances to the nearest pore grid cell. Because all the neighbors of the dark blue grid cell are saturated with biomass, the excess biomass is first moved to a neighboring cell randomly chosen amongst those with the shorter Manhattan distance. This allocation process is repeated until the excess biomass is moved to the grid cell with the Manhattan distance of 1 (color-coded white) where the excess biomass first encounters unsaturated neighbor cells. The biomass holding capacity ($B_{max} - B$) of the randomly selected neighboring grid cell (outlined green) is first evaluated. If the excess biomass exceeds the holding capacity, the excess biomass is redistributed to the first chosen neighbor up to the maximum holding capacity and the remaining excess biomass is placed to the next available neighbor grid cell (outlined yellow), finalizing the cellular automaton algorithm. The horizontal black dashed lines indicate the minimum biomass level required for a grid cell to be designated as biofilm.

When the sessile biomass reaches a threshold density ($B \geq XB_{max}$, where $X$ is a user-defined threshold biomass fraction; $0 \leq X \leq 1$) the pore grid cell is designated as biomass; if the biomass density falls below the threshold due to microbial decay or



detachment ($B < XB_{max}$), then a biomass grid cell is converted back to a pore. Pore grid cells allow for both advective and

diffusive transport, but in the biofilm, sessile biomass hinders (i.e., permeable biofilm) or prevents (i.e., impermeable biofilm)

flow and the feedback between biomass growth/decay and advective flow conditions is accounted for by rerunning the flow

solver to steady state after updating biomass distribution and corresponding pore geometry (Jung and Meile 2021). The reduced

advective transport efficiency in permeable biomass grid cells is implemented by modifying local fluid viscosity while for

impermeable biomass, a bounce-back condition is imposed (Pintelon et al., 2012). After imposing the new steady-state flow

field, a streaming step of the transport solver is executed (Figure 1).

## 4 Model verification

To verify the *CompLaB* implementation, the engineered metabolic interaction between two   co-dependent mutant strains (*E.

coli* K12 and *S. enterica* LT2) originally established by Harcombe (2010) and implemented in COMETS (Harcombe et al.,

2014) and IndiMeSH (Borer et al., 2019) was chosen as a test case. *E. coli* K12 is deficient in producing methionine and relies

on the release of methionine by the mutant *S. enterica* LT2. In turn, *S. enterica* LT2 requires acetate released by *E. coli* K12

because of its inability to metabolize lactose. As a result, these genetically engineered strains are obliged to engage in mutual

interaction where neither species can grow in isolation. The ratio of the two strains converged to a stable relative composition

after 48 hours in all the *in vitro* and *in silico* experiments in both initial ratios of 1:99 and 99:1.

Both COMETS and IndiMeSH integrate flux balance cell models of these two microorganisms into a two-dimensional

environment in which metabolites are exchanged via diffusion. The conditions of these simulations were mirrored in *CompLaB*,

with 100 grid cells containing $3 \times 10^{-7}$ g biomass each (total = $3 \times 10^{-5}$ g biomass) distributed randomly across a two-dimensional

domain of 25×25 grid squares. The grid length was set to 500 μm, and the initial distributions of the two species were allowed

to overlap. Three replicate simulations were carried out for each initial microbial ratios of *E. coli* and *S. enterica* (1:99 and

99:1). For the exchange metabolites acetate and methionine, fixed boundary concentrations of 0 mM were imposed on the left

and right side of the domain, respectively, with no flux conditions on the top and bottom boundaries. The concentrations of

lactose (2.92 mM) and oxygen (0.25 mM) were imposed at all domain boundaries. Solute and biomass diffusion coefficients

were fixed at $5 \times 10^{-10}$ and $3 \times 10^{-13}$ m²/s, respectively. Consistent with the IndiMeSH model implementation, the simulation was

carried out using Equation 8 and reduced metabolic models of *E. coli* K12 and *S. enterica* LT2 in which the number of

metabolites and reactions of the original metabolic models had been systematically reduced by an order of magnitude (Borer

et al., 2019).

Figure 3 illustrates the average ratio of *E. coli* and *S. enterica* of all 6 simulations (triplicates for each initial composition

ratios of 1:99 and 99:1) after 48 hours of simulation time. *CompLaB* simulation results agree with both the observations and

the model results of COMETS and IndiMeSH, demonstrating the metabolic inter-dependence of two strains, and the

convergence to a stable composition ratio.





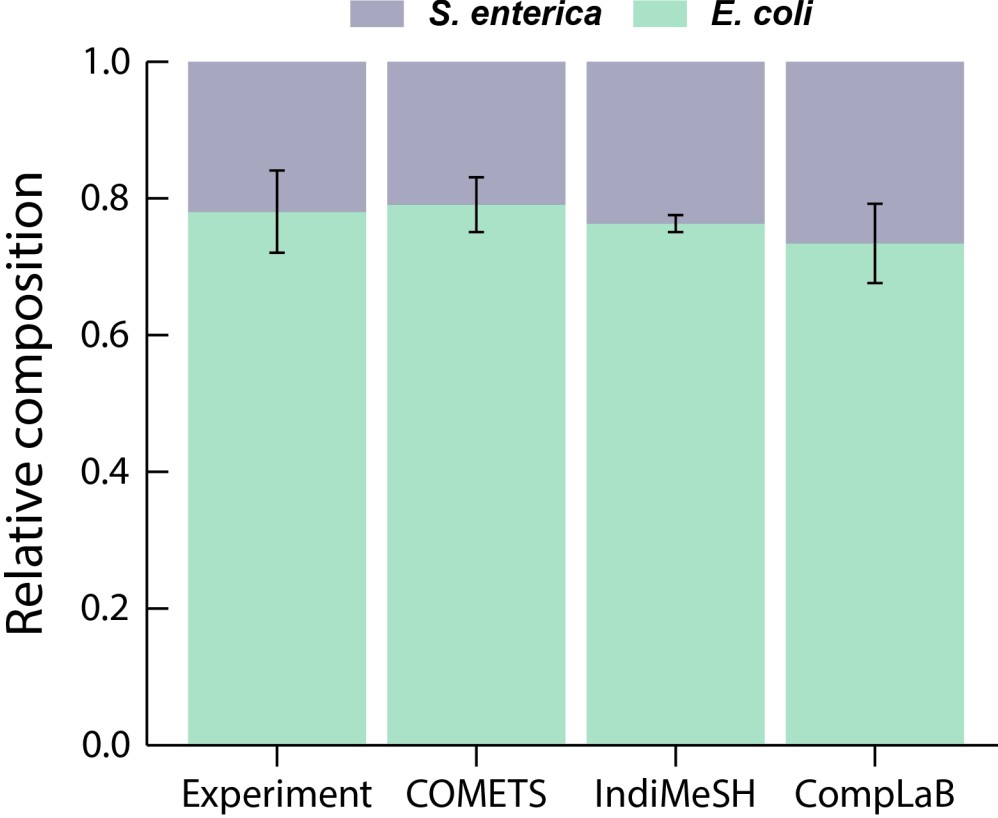

**Figure 3.** The relative composition of 6 in vitro experiments, and COMETS, IndiMeSH and *CompLaB* simulation results after 48 hours. Error bars represent 1 standard deviation.

**5 Surrogate model integration**

A major issue in fully coupling genome-scale metabolic networks to reactive transport models is the large computational demand due to the repeated calculation of the LP problems at every biomass grid cell and every time step. Previous studies have alleviated this issue by interpolating the solutions of LP problems from a look-up table generated in advance to a reactive transport simulation (Scheibe et al., 2009), dynamically creating a solution pool of the LP problems during the reactive transport process (Fang et al., 2011), or systematically reducing the size of the matrix encoding the metabolic network (Ataman et al., 2017; Ataman and Hatzimanikatis, 2017). Here, we introduce a statistical surrogate modeling approach using a pre-trained artificial neural network (ANN) following the approach presented in Song et al. (2022). A trained ANN model directly relates input parameters (i.e., uptake rates of substrates) to outputs (i.e., biomass and metabolite production rates) through a set of nonlinear algebraic equations. As computing such input-output relationships from a pre-trained ANN model is several orders of magnitude faster than running FBA using a full-fledged metabolic model, the use of such surrogate models to achieve a significant speedup has attracted much attention recently (e.g., De Lucia and Kühn. 2021; Prasianakis et al., 2020).



Here, we use the metabolic network of *Geobacter metallireducens* GS-15 (iAF987; Lovley et al., 1993; http://bigg.ucsd.edu/models/iAF987), a strict anaerobe capable of coupling the oxidation of organic compounds to the reduction of metals such as iron and manganese, and using ammonium as nitrogen source to train a ANN model. The dataset used for training a surrogate ANN model was obtained by collecting FBA solutions of the base model obtained from `cplex` with the objective function chosen to maximize biomass production. The solution set was prepared by randomly sampling

5,000 combinations of two growth-limiting metabolites, acetate ($C_{Ac}$) and ammonium ($C_{NH_4}$), within the concentration ranges of $0 \leq C_{Ac} \leq 0.5$ mM and $0 \leq C_{NH_4} \leq 0.05$ mM via Monte-Carlo simulations. Substrate concentrations were converted to uptake fluxes via the parameter-based approach (Eq. 7) using the parameters from Fang et al. (2011) ($V_{max} = 10$ mmol acetate/g$_{DW}$/h and 0.5 mmol ammonium/g$_{DW}$/h; $K_s = 0.01$ mM acetate and 0.1 mM ammonium). These fluxes were used as lower bounds and $Fe^{3+}$ was allowed to be consumed without limitation. The collected FBA solution dataset was split and used to train (70%),

validate (15%), and test (15%) and we developed an ANN model using MATLAB's neural network toolbox. As key hyperparameters, the number of layers and the number of nodes in the ANN model were respectively determined to be four and ten through grid search. Ensuring the accuracy of a surrogate model is of critical importance in reactive transport models because even small errors accumulating over successive timesteps can lead to a substantial error. In this simple example, the trained ANN estimates the biomass growth rate of the full FBA model almost perfectly (R > 0.999) against training, validation,

and test datasets. This shows that the full-fledged metabolic model can be replaced by the surrogate ANN model without substantial loss of accuracy, boosting the simulation speed as shown next.

## 6 Model performance

    *CompLaB* inherits the massive scalability of Palabos which decomposes a simulation domain into multiple subdomains and assigns them to individual computational nodes. In the following, the scalability of various components of *CompLaB* is

assessed for a simplified microbial dynamics problem.

### 6.1 Test case

    The simulation domain (Figure 4) was prepared by taking a subsection of $500 \times 300$ square elements from the porous medium of Souzy et al. (2020). The length of each element was 16.81 μm, and material numbers were assigned to solid (0), pore (1), solid-pore interfaces (solid side of interface = 2, pore side of interface = 3). Ten percent of the interface grid cells (pore side)

were randomly assigned as sessile biomass grid cells (4) initially. Flow was induced from left to right by imposing a fixed pressure gradient between left- and right-side boundaries and no flow conditions were set on the top and bottom boundaries, as well as on the grain surfaces. The steady-state flow field was then provided to the *CompLaB* transport solver for mass transport and reaction simulations (*Péclet number* = 1). Two growth limiting metabolites, acetate ($CH_3COO^-$) and ammonium ($NH_4^+$), were considered for the mass transport simulations. Acetate was injected at the inlet (left) boundary with the fixed

concentration of 0.45 mM to the simulation domain initially filled with the same concentration. Ammonium concentration in



the inflowing fluid and initially in the domain was 0 mM, but it was produced at solid surfaces assuming a zeroth-order mineralization rate (Table 1). For both metabolites, no gradient conditions were imposed at the outlet, top/bottom, and grain boundaries. No external source and initial planktonic biomass were assumed, so that all planktonic biomass is detached sessile biomass.


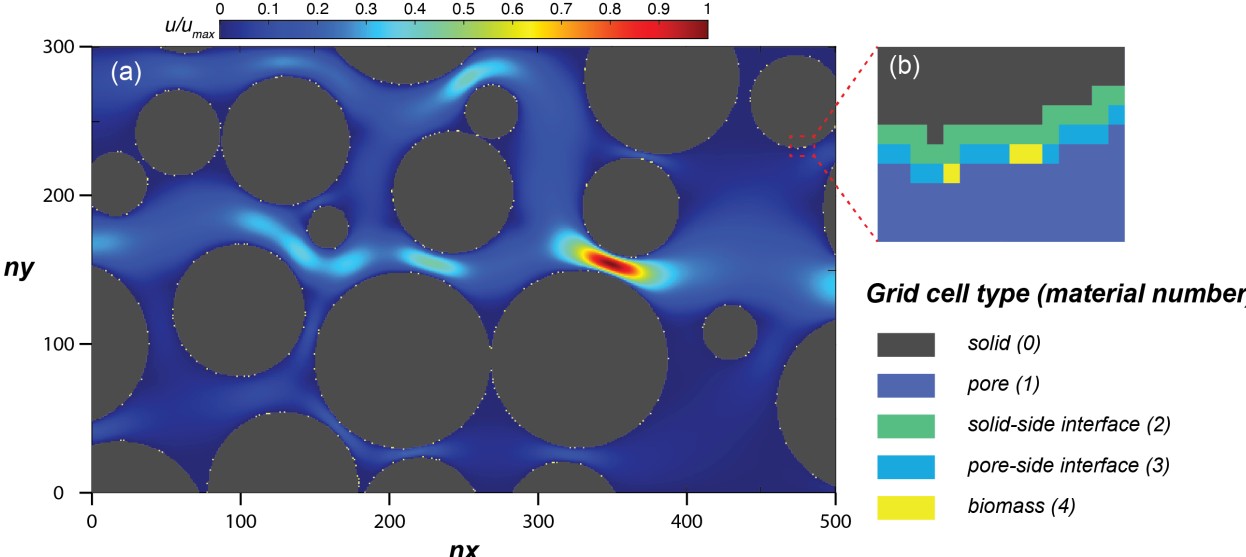

**Figure 4.** (a) Simulation domain of the model performance simulation, with the steady-state flow velocity distribution and (b) an enlarged subsection showing the distribution of material numbers around a solid.

The biogeochemical problem is described by the following set of ADREs:


$$\frac{\partial C_{Ac}}{\partial t} = \nabla \cdot (D_{Ac}\nabla C_{Ac}) - \mathbf{u} \cdot \nabla C_{Ac} - F_{Ac}B_s \tag{9}$$

$$\frac{\partial C_{NH_4}}{\partial t} = \nabla \cdot \left(D_{NH_4}\nabla C_{NH_4}\right) - \mathbf{u} \cdot \nabla C_{NH_4} + k_{NH_4}\delta - F_{NH_4}B_s \tag{10}$$

$$\frac{\partial B_P}{\partial t} = \nabla \cdot \left(D_{B_p}\nabla B_p\right) - \mathbf{u} \cdot \nabla B_p - \mu_B B_p - k_{att}B_p + k_{det}B_s \tag{11}$$

$$\frac{\partial B_S}{\partial t} = \gamma B_s - \mu_B B_s + k_{att}B_p - k_{det}B_s \tag{12}$$

$C_{Ac}$ and $C_{NH_4}$ are the concentrations of acetate and ammonium, respectively, $B_P$ is the planktonic biomass density, $B_S$ is the sessile biomass density, $\mathbf{u}$ denotes the flow field, $\delta$ indicates the presence ($\delta = 1$) or absence ($\delta = 0$) of a grain surface, $D_{Ac}$, $D_{NH_4}$, and $D_B$ are the diffusion coefficients of acetate, ammonium, and planktonic biomass, respectively, which differ between

pore, biomass and solids. For simplicity, the diffusivities of all the metabolites and planktonic biomass were set to $10^{-9}$ m$^2$/s in





the pores, $8\times10^{-10}$ m²/s in biomass grid cells, and 0 in the solid. Biomass attachment/detachment ($k_{att}$ and $k_{det}$), biomass decay ($\mu_B$), and organic matter mineralization ($k_{NH_4}$) were simulated using the reaction kinetics solver, with the corresponding rate constants summarized in Table 1. The simulation assumed that *G. metallireducens* grows only on solid surfaces.


**Table 1.** Parameters used for all the model performance simulations.

| Symbol | Description | Value | Unit | Source |
|--------|-------------|-------|------|--------|
| $k_{att}$ | Biomass attachment rate constant | $2.1 \times 10^{-3}$ | 1/s | King et al., 2010 |
| $k_{det}$ | Biomass detachment rate constant | $1.2 \times 10^{-5}$ | 1/s | King et al., 2010 |
| $\mu_B$ | Biomass decay rate constant | $1.74 \times 10^{-6}$ | 1/s | Fang et al., 2012 |
| $k_{NH_4}$ | N mineralization rate constant | $6.2 \times 10^{-5}$ | mM/s | Trinsoutrot et al., 2000 |

The computational efficiency and parallel performance of *CompLaB* was tested by executing four independent simulations
utilizing each reaction solver. The cell-specific metabolic fluxes ($F$, Eq. 9) and biomass growth rates ($\gamma$, Eq. 12) were calculated either through FBA (CPY or GLPK), ANN, and/or a reaction kinetics (KNS). The KNS solver was combined with other solvers (CPY, GLPK, and ANN) or used as a stand-alone reaction solver with the cellular automaton algorithm invoked (CA). The model performance simulations with the FBA solvers (CPY and GLPK) were prepared with the same conditions used for training the ANN model (section 5). The pretrained ANN model from section 5 was used for the separate simulation ANN.
For the CA simulation, we create a situation similar to the above examples, but in which substantial biofilm growth over a short simulation time is artificially induced. To that end, $F$ and $\gamma$ were computed as:

$$F = k_{kns}\left(\frac{C_{Ac}}{C_{Ac} + K_{Ac}}\right)\left(\frac{C_{NH_4}}{C_{NH_4} + K_{NH_4}}\right) \tag{13}$$

$$\gamma = YF_{Ac} \tag{14}$$

where $F$ denotes the metabolic fluxes (for simplicity assuming $F_{Ac} = F_{NH4}$), $\gamma$ is the biomass growth rate, $k_{kns} = 2.5 \times 10^{-6}$ mM/s
(Marozava et al., 2014) is the maximum uptake rate, and $K_{ac} = 0.1$ mM and $K_{NH_4} = 0.01$ mM are the half saturation constants for acetate and ammonium, respectively. The growth yield $Y$ is set to 40,000 $g_{DW}$/mmol_$Ac$, an arbitrarily large number used only to invoke the CA algorithm within 10,000 timesteps. The flow field was updated every 10 timesteps when the CA algorithm was invoked.

The performance tests were carried out on the computing nodes of Georgia Advanced Computing Resources Center. Each
node has an AMD EPYC 7702P 64-core processors with a 2.0 GHz clock cycle (AMD Rome), with 128 GB of RAM. The





nodes are interconnected via an EDR InfiniBand network, with 100 GB/s effective throughput and run a 64-bit Linux operating system (CentOS 7.9 distribution). The elapsed (wall-clock) time for 10,000 timesteps was recorded.

## 6.2 Performance

Comparison of simulation times for flow, transport and reaction shows that most compute time is used for simulating the reactions, in particular when integrating *in silico* cell models into a reactive transport framework (Figure 5). This highlights the benefit of using efficient surrogate models. The surrogate ANN model substantially improves computational efficiency compared to CPY and GLPK (about 2 orders of magnitude in total elapsed time; Figure 5a) because calculating the pretrained ANN is much faster than solving the linear programming problem every time step in FBA (Figure 5c). The ANN simulation even exhibits comparable but slightly shorter simulation times than the traditional reaction kinetics calculation (KNS) because of the ANN implementation only operates on biomass grid cells while KNS operates both biomass and pore grid cells.

In addition to the computational efficiency, negligible errors introduced by the surrogate ANN model assure the use of a surrogate ANN model (Figure A1). Although the errors in biomass calculation accumulates over simulation timesteps, it is kept to very low values throughout the simulation (on the order of $10^{-9}$; Appendix A) and has practically no influence on metabolite concentration calculations. This observation illustrates that *CompLaB* can calculate microbial metabolic reactions in heterogeneous porous media based on the genome-scale metabolic model with the superior computational efficiency of a surrogate model without losing accuracy.

The computational efficiency of ANN also works in favor of scalability. The reaction processes of *CompLaB* are inherently an embarrassing parallel task because calculating biogeochemical subprocesses is completely independent of the neighbors (except CA) and all model performance simulations show the reasonable scalability up to 64 cores (Figure 5). However, the scaling behaviors of all simulations except ANN illustrate suboptimal scalability with no or limited speed-up when using more compute resources. The loss of efficiency originates mostly from the calculation of reaction processes (Figure 5c) because the domain decomposition applied to the heterogeneous simulation domain (Figure 4) resulted in an uneven distribution of biofilm grid cells per subdomain and hence a variable size of the problem to be solved on each core. In fact, in our simple example problem (total $500 \times 300$ computational grid cells, constant random seeds), domain decomposition when using 64, 128, and 192 cores produced 6, 38, and 76 subdomains, respectively, that have no initial biomass grid cells. Such subdomains do not contribute to computing microbial metabolisms (FBA) and biomass redistribution (CA), preventing the ideal parallel performance (Figure 5d). While this is also true for ANN, computational efficiency of ANN reduces the time wasted by such subdomains. As a result, the suboptimality is not readily apparent in ANN (Figure 5c).

The CA algorithm implemented in *CompLaB* is a nonlocal process requiring information of neighboring grid cells, but the result exhibits still a good scalability up to 64 cores and suboptimal scalability with more cores used like the other simulations. The CA simulation required less time than FBA simulations because the metabolic reactions were calculated using KNS. But CA spent extra time that was not used by other simulations in updating flow field (Figure 5b) and redistributing excess biomass (Figure 5c-d). This illustrates that the actual time required for the CA algorithm depends on the nature of the biomass expansion.



For example, more time will be required for a system with rapid biofilm growth excess because excess biomass has to travel
a longer distance through a thick biofilm. Furthermore, flow fields will need to be updated more often to reflect the influence
of rapid biofilm growth on flow.



**Figure 5.** The wall-clock time recorded in seconds for (a) total, (b) flow and transport, (c) all reactions, and (d) each reaction part of the
*CompLaB* algorithm. Four different simulations were carried out deploying each reaction solver: Three simulations used COBRApy (CPY),
GLPK, artificial neural network (ANN) solvers for microbial metabolism calculation and a kinetic solver (KNS) for the
attachment/detachment and decay of planktonic biomass (AT/DT&DC). One simulation used only a kinetic solver for both microbial
metabolism and AT/DT&DC with the cellular automaton algorithm invoked (CA). Each symbol represents the average of 2 simulations.
The simulations exhibiting no or limited speed-up with 128 and 192 cores were excluded from drawing the regression lines except the ANN
simulation. The negative numbers are the slopes of the solid regression lines. The average slope of all the dashed regression lines in panel
(d) is -1.14.



## 7 Conclusions

The numerical modeling platform *CompLaB* for simulating 2D pore scale reactive transport processes is capable of utilizing quantitative implementation of microbial metabolism through the coupling of genome-scale metabolic models. The integration
of *in silico* cell models with reactive transport simulations makes this framework broadly applicable and enables the integration of knowledge gained from the omics-based characterization of microbial systems. For example, the successful reproduction of experimentally observed convergences to a stable composition of a two species consortium (*S. enterica* and *E. coli*) demonstrates the capability of *CompLaB* to investigate metabolic interactions between multiple microbial species.

Our novel numerical framework based on the Lattice-Boltzmann method allows simulating advection as well as diffusion
of metabolites in complex porous media. A wide range of simulation domains can be used to represent soil structure and fractured rock images directly obtained from various imaging techniques (e.g., μ-CT, FIB-SEM, etc.) or numerically generated porous media with material numbers assigned to pore, solid, and source/sink grid cells for biogeochemical reactions which include but are not limited to biofilms. The inherent parallel efficiency of *CompLaB* facilitates simulating dynamic flux balance analysis capturing the microbial feedback on flow and transport in porous media. Furthermore, the versatile simulation
environment of *CompLaB* allows utilizing surrogate models, such as an artificial neural network. The resulting speed-up enables the investigation of complex biogeochemical processes in natural environments.

## Appendix A. Surrogate model accuracy

Surrogate modeling approach inevitably introduces errors in model estimations. The errors should be maintained sufficiently low throughout the surrogate simulation otherwise errors can propagate in successive iterations and result in unphysical results.
To quantify the errors in surrogate model estimations, the solutions of our artificial neural network (ANN) were compared to the reference simulation COBRApy (CPY) by calculating the arithmetic mean of the root mean squared errors:

$$error_t = \frac{1}{m} \sum_{j}^{m} \frac{\sqrt{\frac{1}{n_j} \sum_{i}^{n_j} \left( CPY_{i,j} - ANN_{i,j} \right)^2}}{\max \left( CPY_j \right)_t} \tag{A1}$$

where $j$ is the variable type ($B_S$, $B_P$, $C_{Ac}$, and $C_{NH_4}$), $m$ is the number of variables $j$ used in calculating the error, $n_j$ is the number of grid cells for each variable $j$, and $t$ is the timestep where the error is evaluated. The differences between the FBA simulations using GLPK and CPY solvers are negligible, so only the CPY solution was chosen as the reference (Figure A1).



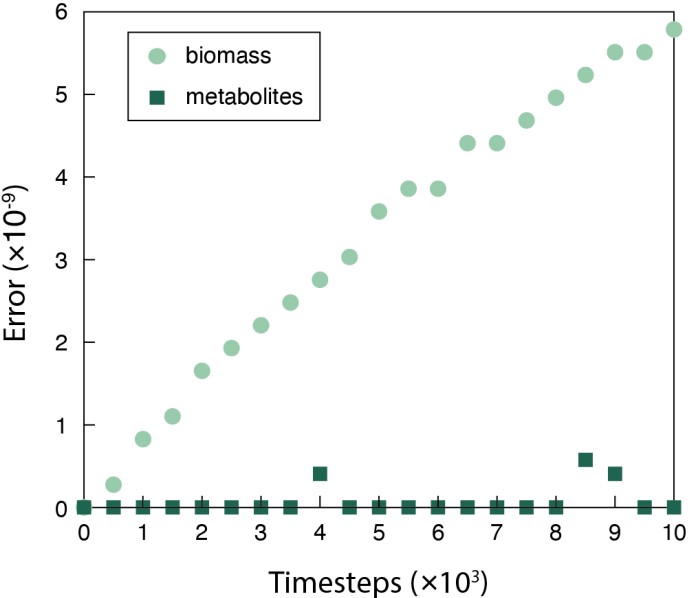

**Figure A1.** The discrepancy between the surrogate ANN and the full-fledged physical model simulation using COBRApy calculated via Equation A1. The surrogate model overestimates biomass ($B_S + B_P$) and the errors accumulate over time. But the errors are kept low and negligible throughout the simulation, as evidenced by no influence on metabolite concentrations ($C_{Ac} + C_{NH_4}$).

**Code availability.** The model code, input files used for this study and a manual are available at https://doi.org/10.5281/zenodo.7095756. Developments after publication of this article will continue to be hosted at https://bitbucket.org/MeileLab/complab/.

**Author contributions.** HJ developed the research, performed the overall programming and simulations, analyzed and interpreted the data, wrote the initial draft and revised the manuscript, HS trained the ANN model and reviewed the manuscript,
CM conceived the research, carried out the performance measures, and revised the manuscript.

**Competing interests.** The authors declare no competing interests.

**Financial support.** This work was supported by the U.S. Department of Energy, Office of Science, Office of Biological and Environmental Research, Genomic Sciences Program under Award Number DE- SC0016469 and DE-SC0020373 to CM, and the Institute for Korea Spent Nuclear Fuel (iKSNF) and National Research Foundation of Korea (NRF) grant funded by
Ministry of Science and ICT (MSIT) under Award Number 2021M2E1A1085202 to HJ.

**Acknowledgements.** The authors thank Shan-Ho Tsai for help with the simulations, and Benjamin Borer for constructive discussions and feedback on IndiMeSH. This study was also supported by resources and technical expertise from the Georgia Advanced Computing Resource Center, a partnership between the University of Georgia's Office of the Vice President for Research and Office of the Vice President for Information Technology.



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
