# Peer review of "CompLaB v1.0: a scalable pore-scale model for flow, biogeochemistry, microbial metabolism, and biofilm dynamics"

_EGUsphere, 2022_

## Author Comment (AC1)

Response to reviews

We thank both reviewers for their valuable feedback and address their comments in detail below. Reviewer comments are in *italics*, our responses in plain text, with changes to the manuscript highlighted in **bold**.

**RC1**: 'Comment on egusphere-2022-1016', Anonymous Referee #1, 09 Nov 2022 reply

*Review of egusphere-2022-1016*

*The manuscript "CompLaB v1.0: a scalable pore scale model for flow, biogeochemistry, microbial metabolism, and biofilm dynamics" by Jung et al. (egusphere-2022-1016) introduces a modular numerical model approach for reactive transport and microbial growth in fully water saturated pore structures. The model can implement/consider high resolution scans of porous media as well as microbial metabolic reaction networks from databases, two growing sources of information on subsurface environment and the processes therein. Model linking this information to reactive transport simulations are still scarce which makes the presented model a potentially very useful tool. The manuscript is well written and introduces the model and its different features. Results on the accuracy and on the model performance are shown.*

*I suggest publication of this manuscript after some moderate revisions. In addition to my comments below, these revisions should clarify which parts of the model have been introduced and verified before and which parts are new and need to be verified in the manuscript (if not done already). I have no worries regarding the technical accuracy of the model but more information on this would be good. This would then also allow determining where the presented model is more advanced than previous models.  I also think it would help to show more results of the presented simulation examples (in the manuscript or in some supplement) to a) demonstrate the model performance and b) to allow putting the discussed results in a better context.*

We appreciate the favorable assessment and have now clarified which parts of the model have been used previously, as outlined in the detailed response below.  We have also added **Figure A1** to the supplement to provide more context for our findings.

*Specific comments:*

*Introduction: I am missing a bit some statements on what exists already for modeling reactive transport and microbial processes at the pore scale There are several rather recent reviews on this (e.g., König et al., 2020, doi: 10.3389/fevo.2020.00053; Golparvar et al., 2021, DOI: 10.1002/vzj2.20087; Pot et al., 2022, DOI: 10.1111/ejss.13142).*

We agree that providing some more context is valuable and have integrated additional information on existing efforts. Specifically, we added at the end of the first paragraph:

**Notably, computational efficiency and the integration of adequate formulations of microbial function has been identified as critical aspects in pore scale models of microbial activity (Golparvar et al., 2021).**

And, following the first sentence in the second paragraph of the introduction:

… biogeochemical processes. **Our work complements existing efforts, encompassing both individual- and population-based spatially explicit microbial models reviewed by König et al. (2020), some of which take into consideration the structure of the porous medium. Our modular** framework is developed …

References cited:

Golparvar, A, Kaestner, M. and Thullner, M. 2021. Pore-scale modeling of microbial activity: What we have and what we need. Vadose Zone Journal;20:e20087.

König, S., Vogel, H.-J., Harms, J. and Worrich, A. 2020. Physical, chemical and biological effects on soil bacterial dynamics in microscale models. Frontiers in Ecology and Evolution. 8:53. doi: 10.3389/fevo.2020.00053

*L 79: Clarify if "based on the LB method implemented in Jung and Meile" means you are using the previously established code and implement the new features or you have code new flow and transport modules based on the same LB concept. This determines which parts of the model need to be verified in this manuscript and for which parts a verification is given already in previous publications.*

This is now clarified by adding the following to the manuscript after the first sentence in the section Model description"

… Jung and Meile (2019, 2021). **These earlier efforts established some of the underlying model developments, such as the simulation of the flow field, mass transport, and biochemical processes including kinetic rate expressions and cellular automata implementation of biofilm growth. This study expands on the previously established models to offer a much broader applicability by building the modular structure that makes the use of flux balance and surrogate models possible.** The LB…

*L 120: How is this combination achieved?*

Clarified as follows:

… or combinations thereof, **by summing their contributions to the net reaction rates of individual state variables.**

*L 124: Clarify if the considered microbial dynamics are limited to specific example processes (and their kinetic expressions) or if any arbitrary (user defined) processes/rate expressions can be used.*

Clarified as follows:

… including Monod kinetics, microbial attachment**/detachment, and arbitrary rate expressions defined by the user.** Reactions …

*Section 3.2.1: Related to my comment above, in case any arbitrary set of rate expressions can be considered the approach is a) not limited to microbially controlled reactions and b) would conceptually not make a difference between the concentration of biomass and chemical compounds. It is thus not clear to me why there is a distinction between these concentrations at this stage. At the end Eq. 5 is just a specific version of Eq. 6 in case of R is given as gamma\*B.*

The reviewer is correct that there is no truly fundamental difference between biomass and chemical compounds per se in our model. We chose to present them separately for two reasons. First, it aligns with the general setup of the manuscript, in which we emphasize the use of the model to deal with microbially-mediated reactions, including the coupling with Flux Balance Models. Second, we have developed a user interface in which microbes are dealt with separately from chemical species due to the link with the above mentioned in silico models, and differences in transport.

Because this decision only reflects the existing setup, we opted to keep it this way. However, we plan to add more examples (with and without microbial dynamics) on our online repositories in the future that will allow users to build on. We now refer to different state variables in the caption of figure 1, where we added: **The state variables involved in the reactions can represent dissolved chemicals and planktonic microbes, and solid phases or sessile microorganisms, respectively.**

*L 148-162: Is there any specific reason why these variables must have the given units?*

In general, arbitrary units can be used but they need to be consistent. In our FBA model simulations, the unit of lower bounds had been set to mmol/gdw/h, hence the choice of units in our CompLab implementation.This is clarified as follows

… a linear function to local concentrations. **Note that the units in the fluid flow and mass conservations model simulations must match those of the FBA bounds, which in our case were mmol/gdw/h.** With lower bounds defined, …

*L 213-220: Were the imposed initial and boundary conditions comparable to the reference models and the experiments?*

Yes. This is now clarified as follows:

Both COMETS **(Harcombe et al., 2014)** and IndiMeSH **(Borer et al., 2019)** integrate … The **initial and boundary conditions** of these simulations were mirrored …

*L 228-229: How is it shown that the composition of the bacteria is stable? Fig. 3 shows that after 48 all approaches exhibit approximately the same composition but not that this composition will not change later on.*

We have now added an appendix (new appendix A), in which we show the composition of the microbial community over time and state the following

**Appendix A. Convergence of the verification model to a stable ratio after 100 hours**

The 6 simulation cases used in section 4 for model verification were run for 100 hours of simulation time to further evaluate if the observed convergence to an average composition ratio is stable (Figure A1). The composition ratio observed after 48 hours (0.75) is largely maintained through the extended simulation period (increases only to 0.78 after 100 hours).

[Figure]

Figure A1. The evolution of the fraction of *E. coli* relative to the total number of cells (E. coli + S. enterica) over 100 hours. Dashed and dotted lines denote an initial abundance of 99 and 1% E. coli, respectively.

*L 251-253: How well could the results of the FBA simulations be fitted by a kinetic approach using a Michaelis-Menten consumption rates with the parameters given here and a constant growth yield fitted to the FBA results?*

We aimed at comparing the performance of different microbial representations, but did not explore this question in our manuscript. In more complex (and more realistic) scenarios, one would expect differences in kinetic vs. FBA approaches (e.g. if there are changes in phenotypes under different environmental conditions). Hence we opted against expanding on this topic in this manuscript.

*L 273: To which length scale does the Peclet number refer to?*

Clarified as:

… reaction simulations (Peclet number = 1**, for a characteristic length scale of 2 mm**). Two growth limiting …

*L 312: What was the time step size?*

The timestep was 0.044 seconds. It is now clarified as

… within 10,000 timesteps **(440 seconds)**.

*Fig. 5 and associated text passages: It would have been interesting to see how the "traditional" KNS approach without the CA performs compared to the ANN approach. This would also show how much additional computation time the CA requires (besides the larger time for the flow (and transport?) simulation. Since most of the shown examples consider steady state conditions for the flow field but transient conditions for the transport I am wondering why only the computation time for flow and transport together is shown.*

We only compared ANN to FBA, but not to KNS. As mentioned above, this is because the FBA/ANN approaches can intrinsically capture different behavior/metabolic expressions under different environmental conditions, while for KNS this is parameterized. Thus, we decided not to further explore this comparison.

*L 323-324: Following my comment above: Is this shown somewhere?*

The KNS simulation time is presented in Figure 5d (orange symbols with dashed lines). Although it was used only for attachment/detachment calculation, there is practically no difference in terms of the computational time to the KNS metabolism calculation because all the KNS approaches are defined by the same C++ template. The calculation time for CA only can be inferred by subtracting the time used for the KNS only case (orange triangle symbols with the green dashed line, Figure 5d) from the KNS including CA case (gray triangle symbols with the green solid line, Figure 5d). Thus, with the above-mentioned reason, we decided not to further explore this comparison. This is now clarified in the manuscript and Figure 5 caption as follows

"… every time step in FBA (Figure 5c). **For reaction calculations (Figure 5d), the ANN simulation (light blue solid line with gray square symbols)** even exhibits … reaction kinetics calculation (KNS**; dashed lines with orange symbols**) because of …

Figure 5 caption: … with the cellular automaton algorithm invoked (CA). **The simulation time for CA only can be inferred by subtracting the time for AT-DT with CA (green dashed line with orange triangle symbols) from the time for KNS including CA (green solid line with gray triangle symbols) in (d)**. Each symbol … "

*L 328 and Figure A1: Clarify that this a relative error for the biomass/metabolites in the system.*

Clarified as:

L 328: … kept to very low values throughout the simulation (**the relative error is** on the order of $10^{-9}$; Appendix A) …

Figure A1 caption: Figure A1. The discrepancy **(relative error)** between the surrogate ANN …

*L 329-331: In this context it would be interesting to know how much computational effort was needed to run the Monte Carlo simulations for training the ANN model. I am aware that quantifying the training effort is not straight forward but some words on this would be helpful.*

Applying Monte-Carlo simulations to generate FBA data to train a neural network model imposes no significant computational burden in our case because it means performing linear programming (LP) 5,000 times with a set of uptake rates of acetate and ammonium randomly chosen by running a matlab function, rand.m. Using the training/validation datasets (excluding the testing dataset), we determined the number of layers and nodes in the neural network to be four and ten, respectively, because no further improvement in model performance was observed beyond those values. We expanded the description of model training in the revision as follows:.

… without losing accuracy. **Applying Monte-Carlo simulations to generate FBA data to train a neural network model required solving the linear programming problem 5,000 times with a set of randomly chose uptake rates of acetate and ammonium, which does not add a significant computational burden. In our application, we determined the number of layers and nodes in the neural network to be four and ten, respectively, because no further improvement in model performance was observed beyond those values.**

*L 340-343: I do not get this argument. If I assume that each methods invest a given time for computation in the biomass cells and an and a given time for computation (wasted) in the non-biomass cells why should they differ in their scaling behavior?*

Qualitatively, the scaling behavior reflects the implementation of the parallel computations. For example, let's consider a domain consisting of 100 total grid cells that includes a single biomass grid cell and the computation takes 10 seconds. As we observed, most of the computation time is consumed by reaction calculations (FBA and CA), so assume 9 seconds for reaction calculations and 1 second for the rest. If this domain is divided into two subdomains consisting of 50 total grid cells each, only one subdomain contains a biomass grid cell. In this case, the core calculating the subdomain with no biomass would just have to wait for the calculation of the other domain, so the time that takes to calculate the 2 subdomains will be approximately 9.5

seconds or so, plus the time of passing information between nodes. Hence, the domain decomposition does no longer speed up the overall computation.

We have clarified this by adding

… biomass redistribution (CA) **that consumes most of the computational cost (e.g., GLPK calculation consumes ~ 99% of the total computational cost),** preventing …

*Furthermore, since detachment and attachment of the biomass is considered there would be biomass also ending on initially uninhabited surfaces. Perhaps it would be good to actually show some results of these simulations and not only the computation times. Similarly to my comment above: How is the scalability of the KNS approach without CA?*

In the test simulations, the attachment was assumed to take place only at the biomass grid cell (which was not spelled out in our original manuscript), which prevents colonization of previously uninhabited regions. We very much appreciate the thorough review and pointing this out and it is now clarified in the manuscript by adding delta_S and delta_B terms in equations 11-13 to indicate the dependence on the presence of solid interface and biomass grid cells, respectively. It is also reflected in the manuscript as follows.

… The simulation assumed that G. metallireducens grows only on solid surfaces **and planktonic biomass attaches only to existing surface-attached aggregates (Grinberg et al., 2019)**.

Reference: Grinberg, M, Orevi, T, and Kashtan, N.: Bacterial surface colonization, preferential attachment and fitness under periodic stress. PLOS Comput. Biol. 15(3), e1006815, doi: 10.1371/journal.pcbi.1006815, 2019

*L 370: "Soil" might not be the best key word here since soils are typically only partially water saturated while the presented code considers fully saturated conditions. Better to use e.g. "porous media".*

The reviewer is correct that we are only dealing with saturated conditions. However, here we simply refer to the structure of soils, rocks, or artificial porous media and we don't think that our wording implies capabilities that do not exist in our modeling framework.

**RC2**: 'Comment on egusphere-2022-1016', Maria De La Fuente Ruiz, 19 Dec 2022 reply

*This paper presents a novel modeling platform CompLaB for simulating 2D pore-scale reactive transport processes while accounting for microbial metabolism and biofilm dynamics. The manuscript is well-written and offers a fair description of the mathematical framework of the model. However, it lacks fundamental information on the physical processes that are modeled. As well as critical conceptual information, such as, which are the species, phases, and reactions considered in the formulation, how are the model parameters defined and which are their effects on the model performance and outcomes, which specific scientific problem/s are willing to be addressed, or which temporal and spatial scale the model can be extended to. Hence, I highly encourage the authors to add a section for a conceptual description of the model (see for instance Section 3 at https://doi.org/10.1029/2011JB008290 or Section 2 at https://doi.org/10.3390/en12112178).*

We thank Dr. De La Fuente Ruiz for the generally positive feedback on our manuscript and appreciate the thoughtful comments.

Our manuscript is aimed at demonstrating the strengths and capabilities of the newly developed numerical model but not at specific scientific problems. Species, phases, and reactions to which the model is applicable depend on the user-specific applications which only need to be governed by the Navier-Stokes and advection-diffusion-reaction equations (retrieved by the Lattice Boltzmann equations). Also, the temporal and spatial scales depend on grid resolution and computational power available. With the title of the paper referring to the pore-scale, we prefer not to constrain it more but hope for broad application by future users.

We have edited section 3 in an effort to clarify some of the points raised by the reviewer

Now it is explicitly mentioned that:

1. *CompLaB* simulates **a fully saturated** 2D fluid flow and solute transport **at the pore scale** based on the LB method …
2. The LB method retrieves the numerical solutions of the Navier-Stokes (NS) **for fluid flow** and advection-diffusion-reaction equations (ADRE) **for solute transport** by solving …

The Lattice Boltzmann equations (eqs. 1-4) are modified with relations to NS and ADRE explicitly spelled out for improved clarity. **Modifications in section 3.1** include the description of:

1. The relationship of relaxation times with viscosity and diffusivity
2. The calculation of flow velocity (**u**) and solute concentration (C) from particles f and g, respectively.
3. The transported entities (which include solute concentrations and planktonic biomass densities).

The model capability of approximating biofilm permeability by modifying fluid viscosity is also explained in section 3.3 as follows:

The reduced advective transport efficiency in permeable biomass grid cells is implemented by modifying local fluid viscosity **in the biofilm ($n_{bf}$) with $n_{bf} = n_f/X$, where $X$ is a user-defined viscosity ratio ($0 \leq X \leq 1$)**, while for impermeable biomass, a bounce-back condition is imposed (Pintelon et al., 2012).

*In addition, although is stated in the text (e.g. paragraph 370, "CompLaB facilitates simulating dynamic flux balance analysis capturing the microbial feedback on flow and transport in porous media") the authors do not seem to explore/present here such feedback, which to me would be one of the strongest points of the model.*

The emphasis of this paper is on the presentation of capabilities and their computational scaling behavior, so we point the reader to other applications instead. We emphasize this now as follows:

The inherent parallel efficiency of CompLaB facilitates simulating dynamic flux balance analysis capturing the microbial feedback on flow and transport in porous media, **as done previously using Monod-type representations of microbial activity (Jung and Meile 2021).**

*Finally, please find below a few comments/suggestions that the authors should address to clarify and improve the manuscript:*

*Figure 1: What do pore geometry changes stand for? Do you mean pore-clogging by biomass growth? That needs a bit more development within the text, especially because the term porosity does not appear at all or is not included in the few equations presented.*

In the current framework, focusing on microbial dynamics, changes in pore geometry are due to the formation or destruction of biofilm (see section 3.3). Ongoing efforts are expanding these capabilities to include dissolution/precipitation reactions. We now clarify this by adding "**Changes in pore geometry are assessed due to biomass changes**" to the figure caption.

*Also, why are the boundary conditions only applied at the end of the simulation?*

This is a misunderstanding. The boundary conditions are applied after every streaming step, not only at the end of the simulation. We agree that the flow chart was misleading and thus modified the flow chart (Figure 1) to avoid such confusion.

*What is the (gdw) unit used in paragraph 150 and thereafter*

Gram dry weight. It is now introduced when first mentioned ($V_{max}$; **e.g.,** mmol/$g_{DW}$/h, **where $g_{DW}$ is gram dry weight**)

*According to paragraph 155, the time step is measured in hours. How quick are the processes modeled here? How is this time step chosen? And, is there any time-step adaptation process to avoid running out or exceeding biomass concentration within a biomass cell (so that mass conservation is reached)?*

The unit is given in hours to be consistent with the units of metabolic models we integrated into our code  (see our response to the first reviewer). The timestep of typical advection-diffusion simulations are 0.1s ~ 0.001s. Timestep is determined based on the Peclet number which relates the average flow velocity of the flow simulation (eq 1) to the relaxation time of mass transport simulation (tau_g, eq 3). We clarify how the timestep is set in section 3.1.

**In solving an advection-diffusion problem, *CompLaB* adjusts the value of tau_g, which controls the length of a timestep, to obtain a user-provided Péclet number ($Pe^j = UL/D^j$), for a given average flow velocity $U$ and a user-provided characteristic length $L$.**

Biomass exceeding the maximum concentration in a grid cell after a reaction timestep is conserved by redistributing them to the neighboring grid cells as described on Line 174 of the original manuscript. For now, CompLaB does not include adaptive timestepping, to avoid challenges associated with mass conservation (Horstmann et al., 2022. J Comput Phys, 462, 111224, https://doi.org/10.1016/j.jcp.2022.111224). In addition, we have not experienced negative biomass densities and metabolite concentrations, largely because of typically slow microbial processes and small timesteps. Therefore, we decided to leave it for a potential future update and not expand on it in this manuscript.

*Paragraph 175: the excess biomass is redistributed here to a randomly selected neighboring grid cell. However, shouldn't that depend on how much concentration of Biomass there is on the neighboring cells, or perhaps influenced by solid diffusion through the biofilm?*

In our implementation, we only consider if a gridcell is designated as biofilm or not (which is decided by a threshold value). We agree that there are many alternative descriptions that one might consider. We acknowledge that this is at best a first order approximation with ample room for improvement and we acknowledge that by stating "**Note that this biomass redistribution method is a simple approximation for biomass density conservation with room for improvement (e.g., Tang and Valocchi, 2013)**". Much of such future development, in our opinion, should be based on applications that are driven by detailed observations, which is well beyond the scope of this work.

Reference: Tang, Y. and Valocchi, A. J.: An improved cellular automaton method to model multispecies biofilms, Water Res, 47, 5729–5742, https://doi.org/10.1016/j.watres.2013.06.055, 2013

*Paragraph 175:  How is Bmax defined? And is its value consistent with changes in the pore space geometry along the simulation?*

Bmax represents a maximum cell density in a grid cell. Thus, it depends on the grid resolution, and the microbial community and environmental setting. We clarify that this is a user-defined quantity (section 3.3):

… filled up to the maximum holding capacity (*Bmax*, **a value defined by the user**) and then …

We also now list the value used in section 6.1

… a stand-alone reaction solver with the cellular automaton algorithm invoked (CA) **and $B_{max}$ set to 100 $g_{DW}$/L**.

*Figure 2: Is fluid flow in pores used to simulate biomass transport in the porous media? or are they considered immobile when the pore is not fully clogged? (check https://doi.org/10.1016/j.csr.2015.04.022)*

In our applications, we often differentiate between sessile and planktonic microbes (see e.g. Jung and Meile 2021), using attachment/detachment formulations, but this choice is ultimately up to the user. We have clarified this now in the Figure 1 caption. "**The state variables involved in the reactions can represent dissolved chemicals and planktonic microbes, and solid phases or sessile microorganisms, respectively.**"

*Figure 2: How can the biofilm in excess be located in a cell where the flow of metabolites is not allowed? As far as I understand dissolved species are only transported through pores, right, and not within biofilm cells? (see general comment for conceptual clarification)*

Biofilm can be considered as either permeable or impermeable based on each user-specific needs. When biofilm is considered permeable, then metabolite transport occurs both by advection and diffusion. In contrast, when biofilm is impermeable, metabolites are transported through diffusion only. This was described at the end of section 3.3 of the original manuscript and edited to clarify as answered in the above first answer to the second reviewer.

*Why in Eq. 9 and thereafter there is no correction for sediment porosity (at least for the non-solid species)? How is the flow field "u" estimated/calculated here?*

Typically, **u** in equation 9 (which is now eq. 10 in the revised manuscript) describes the flow in the pores, but this is a good point when considering flow in permeable biofilm. There, we consider the reduction of pore space to be effectively captured by the parameterization of the flow field, implemented formally by increasing the fluid viscosity in biofilm grid cells. The user can adjust that relationship in complab by adjusting the value of <viscosity_ratio_in_biofilm> keyword, and hence has the option to appropriately adjust the parameterization if there is data available to do so.

*Paragraph 290: Is organic matter mineralization accounted for changes in available pore space?*

No. Our simulations only cover short (< hr) periods. Over these timescales, and under the conditions simulated, changes in the solid phase are minimal and hence not part of the problem formulation (eq. 9-12 in the original manuscript, 10-13 in the revised manuscript).

*Paragraph 290: How is the solid (particle surface characteristics) accounted for describing biofilm attachment/detachment? Which factors are actually controlling this process here?*

Attachment and detachment occur at the biomass grid cells. They are implemented only depending on the presence of the initial biomass and are first order with respect to the attaching/detaching biomass pool (eq. 11 and 12 in the original manuscript, now eq. 12&13)). No other factors are considered in our implementation. This is reflected in the manuscript by spelling out the dependency of these processes on the presence of biomass grid cells in the equations 11 and 12 (delta_B; now eq. 12&13).

*Paragraph 310: Why is the flow field only updated every 10 timesteps for CA?*

This decoupling was conveniently chosen only to reduce the overall computation time and it has been employed in other studies (e.g., Thullner and Baveye, 2008; Jung and Meile, 2021) for the same purpose. In circumstances where fast reactions result in rapid biomass growth, then such an approximation will introduce some error. However, it also should be emphasized that the main purpose of the test simulation is to demonstrate the model capability and its scaling behavior.

We now clarify this by citing the above-mentioned papers

… invoked **(as, e.g., in Thullner and Baveye 2008, Jung and Meile 2021).**

*Paragraph 330: Please specify what heterogeneous porous media means in this context. How is heterogeneity influencing the model outcomes?*

In the modeling context of CompLaB, heterogeneous porous media can refer to various features, including but not limited to the structure of pore geometry, the spatial distribution of biomass, organic matter, and minerals.

We have clarified this as follows:

   This observation illustrates that *CompLaB* can calculate microbial metabolic reactions **in porous media with heterogeneous distribution of pore, biomass, organic matter, and minerals,** based on the genome-scale metabolic model …

*Paragraph 350: Why is the model not accounting for biomass concentration limitations on the diffusion (D) and flow field (u) values?*

Although some papers reported such relationships (e.g. Stewart 2003), these are largely limited to macroscopic observations and there is significant uncertainty, e.g. associated with differences in EPS, biofilm structure etc. (Wimpenny et al., 2010). Thus, we opted to leave the model simple at this stage but we will consider providing a functionality to update flow and diffusion as a function of biomass density in the future release.

Reference:

Stewart, P. S.: Diffusion in Biofilms, J Bacteriol, 185, 1485–1491, https://doi.org/10.1128/JB.185.5.1485-1491.2003, 2003.

Wimpenny, J., Manz, W., and Szewzyk, U.: Heterogeneity in biofilms, FEMS Microbiol Rev, 24, 661–671, https://doi.org/10.1111/j.1574-6976.2000.tb00565.x, 2000.